# Geometrical Stabilities and Electronic Structures of Ru_3_ Clusters on Rutile TiO_2_ for Green Hydrogen Production

**DOI:** 10.3390/nano14050396

**Published:** 2024-02-21

**Authors:** Moteb Alotaibi

**Affiliations:** Department of Physics, College of Science and Humanities in Al-Kharj, Prince Sattam Bin Abdulaziz University, Al-Kharj 11942, Saudi Arabia; mot.alotaibi@psau.edu.sa

**Keywords:** Ru_3_ clusters, rutile TiO_2_ (110), photocatalysis, green hydrogen production, DFT, oxygen vacancy

## Abstract

In response to the vital requirement for renewable energy alternatives, this research delves into the complex interactions between ruthenium (Ru_3_) clusters and rutile titanium dioxide (TiO_2_) (110) interfaces, with the aim of enhancing photocatalytic water splitting processes to produce environmentally friendly hydrogen. As the world shifts away from traditional fossil fuels, this study utilizes the density functional theory (DFT) and the HSE06 hybrid functional to thoroughly assess the geometric and electronic properties of Ru_3_ clusters on rutile TiO_2_ (110) surfaces. Given TiO_2_’s renown role as a photocatalyst and its limitations in visible light absorption, this research investigates the potential of metals like Ru to serve as additional catalysts. The results indicate that the triangular Ru_3_ cluster exhibits exceptional stability and charge transfer effectiveness when loaded on rutile TiO_2_ (110). Under ideal adsorption scenarios, the cluster undergoes oxidation, leading to subsequent changes in the electronic configuration of TiO_2_. Further exploration into TiO_2_ surfaces with defects shows that Ru_3_ clusters influence the creation of oxygen vacancies, resulting in a greater stabilization of TiO_2_ and an increase in the energy required for creating oxygen vacancies. Moreover, the attachment of the Ru_3_ cluster and the creation of oxygen vacancies lead to the emergence of polaronic and hybrid states centered on specific titanium atoms. These states are vital for enhancing the photocatalytic performance of the material within the visible light spectrum. This DFT study provides essential insights into the role of Ru_3_ clusters as potential supplementary catalysts in TiO_2_-based photocatalytic systems, setting the stage for practical experiments and the development of highly efficient photocatalysts for sustainable hydrogen generation. The observed effects on electronic structures and oxygen vacancy generation underscore the intricate relationship between Ru_3_ clusters and TiO_2_ interfaces, offering a valuable direction for future research in the pursuit of clean and sustainable energy solutions.

## 1. Introduction

Photocatalysis is vital in hydrogen generation, symbolizing a renewable and eco-friendly method for energy production. Utilizing light to initiate chemical reactions, this technology is of considerable importance in the realm of renewable energy sources. At the heart of photocatalytic applications is the idea of generating hydrogen sustainably. Hydrogen, recognized for its clean and high-efficiency energy qualities, has traditionally been produced using fossil fuels. Photocatalytic water splitting, however, presents an eco-conscious alternative. This approach was initially proposed by Fujishima and Honda in 1972. They demonstrated water splitting with a TiO_2_ electrode, establishing a foundation for environmentally friendly hydrogen generation [1].

A vital aspect of photocatalysis lies in its capability to harness solar energy, the most plentiful source of renewable energy. The combination of photocatalysis with solar energy, as elucidated by Lewis and Nocera [2], capitalizes on the immense energy of the sun for sustainable hydrogen generation, signifying a notable leap in this domain. The ecological advantages of producing hydrogen through photocatalysis are significant. This method lessens dependency on fossil fuels, which in turn curtails greenhouse gas emissions, aiding in efforts to combat climate change. Turner’s analysis underscores these environmental merits, highlighting the profound impact of photocatalysis in reducing carbon emissions [3]. Recent breakthroughs in photocatalytic materials have been crucial in amplifying hydrogen production efficiency. Innovations in new materials and nanostructures have resulted in improved light capture and charge carrier segregation. Chen et al. [4] have elaborated on how nanostructured photocatalysts contribute to heightened efficiency, emphasizing the progress in nanotechnology for this application. The importance of photocatalysis in hydrogen generation encompasses not only sustainability and environmental benefits but also the utilization of solar energy and strides in photocatalyst material development. This amalgamation of impacts establishes photocatalysis as an integral technology in renewable energy pursuits, with current studies focusing on maximizing efficiency and scalability for real-world use.

TiO_2_ is a pivotal material in photocatalysis, celebrated for its exceptional efficiency and ability to drive photochemical reactions. The prominence of TiO_2_, especially the rutile TiO_2_ (110) surface, in photocatalytic research stems from a range of inherent qualities and the progress it has fostered in renewable energy technologies. TiO_2_ is acclaimed for its potent oxidative capacity, robust chemical stability, non-toxic nature, and affordability [5,6], rendering it an optimal choice for a variety of photocatalytic uses. These attributes are essential for processes such as water splitting [7,8], air cleaning, and the breakdown of pollutants. Within the various forms of TiO_2_, the rutile phase has garnered substantial interest due to its distinct photocatalytic properties [9]. The rutile TiO_2_ (110) surface, especially, is noted for its advantageous electronic configuration and surface reactivity. Diebold [10] highlighted that the rutile (110) surface shows a unique photocatalytic activity, which is attributed to its specific surface atomic structure and the presence of bridging oxygen sites.

The photocatalytic performance of rutile TiO_2_ is notably augmented by its appropriate band gap energy, conducive to the effective absorption of ultraviolet radiation. This feature plays a crucial role in the generation of electron–hole pairs, indispensable in photocatalytic processes. Thompson and Yates [11] observed that the band structure of rutile TiO_2_ promotes efficient charge transfer, a key component in photocatalysis. In addition, rutile TiO_2_ (110) is recognized for its exceptional capability in photocatalytic water splitting, an essential step in hydrogen generation. The surface attributes of rutile TiO_2_, including the presence of active sites and its proficiency in charge separation, significantly bolster the hydrogen evolution reaction. Fujishima et al. [12] have underscored the capacity of rutile TiO_2_ to elevate the efficacy of photocatalytic hydrogen production. Moreover, the adaptability of rutile TiO_2_ (110) is evident in its potential for doping or alteration with different metals and non-metals to customize its photocatalytic characteristics. Such alterations enhance absorption in the visible light spectrum and reduce the recombination of charge carriers. This concept was examined by Maeda [13], who studied the enhancement of TiO_2_’s photocatalytic efficiency in visible light conditions through diverse modifications.

The photocatalytic prowess of rutile TiO_2_ (110) is notably impacted by its surface reactivity. Factors such as surface morphology, including the presence of defects and vacancies, are crucial in defining its interactions with reactants and intermediates. Notably, oxygen vacancies on the rutile (110) surface are known as pivotal active sites for photocatalytic processes. Henderson [14] has highlighted the importance of these surface defects in augmenting rutile’s photocatalytic efficiency. Beyond its inherent qualities, the photocatalytic performance of rutile TiO_2_ (110) can be further tailored and improved through a variety of treatments and doping techniques. Techniques like metal [15] or non-metal doping [16], surface sensitization, and the formation of heterojunctions with other semiconductors [17,18] have been employed as effective methods to broaden light absorption into the visible spectrum and to improve the dynamics of charge carriers.

The contribution of metal clusters, particularly those composed of Ru, in augmenting the photocatalytic efficiency of materials such as TiO_2_, has been a focal point in contemporary scientific investigations. These metal clusters are recognized for significantly boosting the performance and efficiency of photocatalysts [19], opening up novel prospects in renewable energy and environmental cleanup. Ru, being a transition metal, exhibits distinctive electronic and catalytic features that render it exceptionally effective in improving photocatalytic processes. When Ru clusters are applied to the surfaces of photocatalysts like TiO_2_, they can bring about various advantageous effects. A key benefit is the improvement of light absorption capacity, notably within the visible light spectrum. This enhancement is vital since most photocatalysts, TiO_2_ included, predominantly absorb ultraviolet light, which is just a minor component of the solar spectrum. The application of Ru clusters expands the light absorption range, enabling a larger segment of solar energy to be utilized in photocatalytic reactions.

Yu et al. [20] have showcased the proficiency of Ru in boosting the photocatalytic capabilities of TiO_2_. Their research revealed that applying Ru metal clusters notably enhances the hydrogen evolution reaction (HER) even in challenging environments. Additionally, Ru clusters serve as active sites for photocatalytic reactions, thereby accelerating processes like water splitting and pollutant decomposition. This metal plays a pivotal role in the separation and mobilization of photo-induced electron–hole pairs, diminishing recombination occurrences and thus elevating the overall photocatalytic efficiency. Gao et al. [21] further emphasized this aspect by examining the influence of Ru metal atoms in amplifying photocatalytic activity [15]. Furthermore, the integration of these metal clusters can alter the electronic structure of the photocatalyst, leading to improved dynamics of charge carriers. Such modifications are especially advantageous in operations like carbon dioxide reduction and hydrogen generation through water splitting, where efficient electron transfer is essential. Ren et al. [22] have demonstrated how single Ru atoms can be employed to modify the electronic characteristics of photocatalysts, thereby enhancing their performance.

Ru clusters significantly influence the photocatalytic behavior of rutile TiO_2_ by modifying its optical characteristics. In particular, Ru excels in broadening TiO_2_’s light absorption capacity into the visible spectrum. This expansion is facilitated by the band gap reduction in TiO_2_, resulting from the emergence of new energy levels close to the conduction band (CB) or valence band (VB) due to the integration of Ru. The introduction of these new states enables rutile TiO_2_ to harness a wider segment of the solar spectrum, particularly visible light, thus augmenting its photocatalytic efficiency in the presence of solar light. Li et al. [23] have shown that TiO_2_ laden with Ru clusters exhibits enhanced photocatalytic activity, particularly in water splitting, a development attributed to the Ru cluster’s role in promoting effective charge separation. While there is existing research on the enhancements brought by metal clusters in photocatalysts, the distinct interactions and effects of Ru_3_ clusters on both the pristine and reduced TiO_2_ rutile (110) surfaces remain under-explored. Consequently, this research, employing sophisticated DFT techniques, seeks to expand our comprehension of these interactions and their potential in refining photocatalytic processes when exposed to solar radiation.

In our study, we analyze the behavior of Ru_3_ clusters on both pristine and reduced TiO_2_ rutile (110) surfaces, utilizing the principles of DFT. Our approach specifically involves the application of the DFT-D3 method, chosen for its precision in depicting the adsorption phenomena of Ru_3_ clusters on rutile TiO_2_. Furthermore, we implement the HSE06 hybrid functional, devised by Heyd, Scuseria, and Ernzerhof [24], to conduct an in-depth examination of the electronic characteristic, particularly concerning polaron formation on TiO_2_. This functional stands out due to its integration of a portion of exact exchange, offering a more nuanced view of electronic characteristics compared to conventional DFT methodologies. The structure of this article is as follows: Section 2 details the simulation methods used, focusing on ensuring clarity and reproducibility. Section 3 discusses the results of the simulations, exploring the interactions and behaviors within the system, with a special emphasis on the concept of polaron. This section also compares these new findings with prior research, enhancing the theoretical comprehension of charge carriers on TiO_2_ surfaces. Section 4 summarizes the key findings, underscoring their significance to the wider scientific community, particularly in the field of sustainable energy technologies.

## 2. Computational Details

In the field of computational nanomaterials science, comprehensively characterizing and understanding the electronic attributes and photon absorption capacities of Ru_3_ clusters is a formidable task. This research confronts this challenge by employing advanced computational simulations, employing a diverse approach to precisely represent and scrutinize these essential characteristics. By integrating DFT with the HSE06 hybrid functional, this study investigates the electronic structure of Ru_3_ clusters. The objective of these simulations is to garner a detailed insight into the quantum mechanical interactions within the cluster, which in turn sheds light on the contribution of each atom to the cluster’s overall electronic behavior. Moreover, this research examines the potential application of Ru_3_ clusters in photocatalysis, focusing on how their distinct electronic properties could potentially improve the light absorption efficiency of rutile TiO_2_.

The foundation of the simulation approach in this study is the application of the Vienna Ab initio Simulation Package (VASP 5.4.4) [25,26,27,28], utilizing the HSE06 hybrid exchange–correlation functional. Renowned for its accuracy, this functional incorporates both the short- and long-range elements of the Perdew–Burke–Ernzerhof (PBE) [29] exchange functional, along with a short-range Hartree–Fock (HF) exchange. This blend ensures a comprehensive and precise analysis of electron exchange and correlation phenomena. Additionally, the projector augmented wave (PAW) method [30,31] and PAW-PBE pseudopotentials are applied to intricately define the interactions between ion cores and valence electrons, a key factor in ascertaining the electronic properties of the clusters. The atomic orbitals of Ti, O, and Ru are considered valence electrons, affording an intricate depiction of the electronic milieu. To address the limitations commonly associated with standard DFT methods, especially the self-interaction error that can cause artificial electron delocalization, a generalized-gradient approximation (GGA) augmented with a Hubbard term (U) is utilized. The U value for the 3D orbitals of titanium is set to 4.2 eV, consistent with values found in existing studies [32,33,34].

This research includes a comprehensive modeling of the pristine rutile TiO_2_ (110) surface, which is fundamental for grasping the interactions between Ru_3_ clusters and this specific surface. The depiction of the rutile surface is achieved using a unit cell with defined dimensions, including a 20 Å vacuum layer, which accurately reflects the surface structure typically observed in experimental settings. To effectively model an isolated Ru_3_ cluster, large supercells are employed, specifically sized at 30 Å^3^. This approach is crucial to prevent any unintended interactions with periodic images, an essential factor for precise energy calculations. The simulation parameters are meticulously selected to ensure a balance of computational efficiency and accuracy. These include using a single k-point value, setting the plane waves basis set cut-off energy at 500 eV, and applying a Gaussian smearing of 0.05 eV for band occupation. The self-consistent electronic minimization procedure, adhering to a convergence threshold of 10^−4^ eV and a relaxation force threshold of 0.02 eV/Å, ensures both the stability and accuracy of the simulated structures.

Incorporating van der Waals (vdW) corrections [35] through the spin-polarized Perdew–Burke–Ernzerhof method, in conjunction with the Becke–Jonson damping function [36], is a pivotal element in addressing the complexities of metal–oxide interactions. This step goes beyond being a mere computational aspect; it is vital for accurately capturing the subtle physicochemical interactions that play a vital role in influencing the stability and reactivity of nanoscale materials. The process of computing the adsorption energy ($E_{ads}$) of Ru_3_ clusters, which involves thorough energy considerations, is instrumental not just in assessing the adsorption stability but also in shedding light on the potential catalytic applications of these clusters. The adsorption stability of the Ru_3_ cluster is quantitatively calculated by computing its adsorption energy ($E_{ads}$) using the following established equation:(1)$$E_{ads}=E_{tot}-E_{TiO2}-E_{Ru3}$$
where $E_{tot}$ denotes the total energy of the combined system, $E_{TiO2}$ represents the total energy of TiO_2_, and $E_{Ru3}$ is the total energy of the Ru_3_ clusters. Additionally, the energy associated with the formation of oxygen vacancies ($E_{Vo}$) was calculated using the following equation:(2)$$E_{Vo}=E_{surface+Vo}+\frac{1}{2}E_{O2}-E_{surface}$$

In this formula, $E_{surface+Vo}$ corresponds to the final energy of the TiO_2_ with reduced oxygen, $E_{O2}$ is the final energy of molecular oxygen in its gaseous state, and $E_{surface}$ is the final energy of the pristine TiO_2_. The development and graphical representation of the structures outlined in this study were accomplished through the utilization of VESTA [37].

## 3. Results and Discussion

### 3.1. Isolated Ru_3_ Cluster

The results presented in Figure 1a,b provide significant insights into the geometrical and electronic properties of the Ru_3_ cluster in a gaseous state. The optimized geometry of this cluster, forming a triangular configuration in a doublet state, is indicative of distinct stability characteristics and electronic behaviors. The triangle shape of the Ru_3_ cluster is found to be the most stable Ru3 cluster [38]. Firstly, the stability of the Ru_3_ cluster is critical. The Ru_3_ cluster demonstrates stability in the gas phase, as evidenced by a total energy value of −14.70 eV. The geometrical analysis of the Ru_3_ cluster reveals inequivalence among the bond lengths (d_1_, d_2_, and d_3_), which are 2.20 Å, 2.33 Å, and 2.48 Å, respectively (see Table 1). This inequivalence in bond lengths within the Ru_3_ cluster might contribute to its higher stability, possibly due to the resulting electronic distribution and geometric arrangement.

The density of states analysis further elucidates the electronic attributes of this cluster. The calculated band gap for the Ru_3_ cluster is 1.70 eV. Furthermore, the Bader charge analysis provides essential insights into the electron charge distribution within this cluster. The Ru_1_ and Ru_3_ atoms possess electron charges of −0.001 e^−^ and −0.01 e^−^, respectively, while the Ru_2_ atom has a charge of 0.01 e^−^. This asymmetry in charge distribution could be a contributing factor to the cluster’s observed stability and electronic properties. These factors are crucial in understanding the behavior of this cluster in various applications, particularly in catalysis and material science.

### 3.2. Ru_3_ Clusters Loaded on Perfect TiO_2_

The computation of the electronic densities of states for perfect rutile TiO_2_ (110) surface, yielding a band gap estimation of roughly 3.15 eV (see Appendix A), is a pivotal result. This value closely aligns with previous experimental findings [39]. The expansion of this study to include the geometrical and electronic characteristics of Ru_3_ clusters loaded on both pristine and defective TiO_2_ rutile (110) surfaces marks an important step in understanding surface–cluster interactions. The detailed computational modeling of three distinct adsorption configurations for triangular Ru_3_ clusters reveals how orientation affects cluster stability and interaction with the TiO_2_ surface. The observation that all Ru_3_ clusters, irrespective of their orientation, show no distortions upon adsorption indicates a strong and stable interaction with the TiO_2_ surface. Particularly notable is the finding that the upstanding Ru_3_ cluster configuration exhibits higher stability compared to the tilted (parallel to the bridging oxygen atoms shown in Figure 2b) configurations, with a stability difference of about 0.04 eV. This contrasts with previous studies on Ag_5_ and Rh_5_ clusters, where tilted configurations were found to be more stable [40,41]. This suggests unique interaction dynamics between Ru_3_ clusters and TiO_2_ surfaces, differing fundamentally from those observed in other metal clusters like Ag_5_ and Rh_5_.

Furthermore, the Ru_3_ clusters in the perpendicular orientation to the bridging oxygen atoms (see Figure 2c) showed significantly higher adsorption energy (−5.15 eV), indicating enhanced stability over both upstanding and tilted configurations. This improved stability, evidenced by the mean Ru-O bond distance being around 2.20 Å, indicates a stronger interaction when the cluster aligns vertically to the surface. This insight is vital for comprehending the determinants of adsorption and stability of metallic clusters on oxide substrates, with direct relevance to catalytic processes and materials engineering. Table 2 offers a detailed comparative evaluation, outlining the adsorption energies and corresponding charges for the different configurations of the Ru_3_ cluster, as depicted in Figure 2.

The observed charge transfer of approximately +0.75 e^−^ from the Ru_3_ cluster to the TiO_2_ surface, in the most stable configuration, indicates the induction of an oxidation state in the Ru_3_ cluster. This electron transfer corroborates with previous research [20,21,42,43] and is a fundamental aspect in understanding the interaction dynamics between the cluster and the surface. The utilization of the HSE06 functional for density of states calculations, coupled with wavefunction computations for the structure in Figure 2c (illustrated in Figure 3), has led to significant findings. Notably, the integration of a Ru_3_ cluster onto the TiO_2_ rutile (110) surface introduces mid-gap states within the band gap. The emergence of these mid-gap states is an important factor in altering the electronic structure of the TiO_2_ surface, potentially affecting its photocatalytic properties.

The positioning of the highest occupied molecular orbital (HOMO) of the Ru_3_ cluster at a high-energy level (−0.23 eV, approximately 0.7 eV below the CB edge) further emphasizes the impact of the Ru_3_ cluster on the electronic characteristics of the TiO_2_ surface. It is also observed that a polaronic state (at −0.58 eV) is formed due to the electron gain of Ti_27_ atom of approximately 0.3 e^−^, which is hybridized with the state formed by the Ru orbital. The formation of mid-gap states, and the polaron attributed to the electron transfer from the Ru_3_ cluster, enhances photon absorption capabilities in both the visible and ultraviolet regions [44,45,46]. This is pivotal for photocatalytic applications, as it broadens the range of light that can be utilized in photocatalytic processes. Moreover, the adsorption of the Ru_3_ cluster on the TiO_2_ surface results in the repopulation of the CB, inducing metallic properties within the material system. Such changes in electronic properties have been observed in TiO_2_ systems interfaced with Ag_3_ and Ag_5_ clusters [47]. The positioning of the mid-gap states to accept electrons from the VB under visible light irradiation, owing to the reduced energy separation, facilitates electron transitions that are crucial for enhanced photocatalytic hydrogen production [48].

### 3.3. Ru_3_ Cluster Loaded on Defective TiO_2_

This research furthers our comprehension of the interactions between Ru_3_ clusters and TiO_2_ rutile (110) surfaces, emphasizing the influence of Ru_3_ clusters on the creation of oxygen vacancy, particularly in defective TiO_2_ structures. Utilizing DFT calculations [40], the result confirms that the energy required to create a surface oxygen vacancy on pristine rutile TiO_2_ is about 0.58 eV less than that for a subsurface vacancy. This observation aligns with prior findings [49,50] and is supported by the data in Appendix A. A key aspect of this research involved examining the most stable arrangement of the Ru_3_ cluster on the TiO_2_ surface, as shown in Figure 2c, and assessing its impact on photocatalytic efficiency, particularly in relation to the presence of surface oxygen vacancies. The findings reveal that incorporating the Ru_3_ cluster onto the TiO_2_ rutile (110) surface results in increased stability, evidenced by a 0.38 eV rise in the formation energy for surface oxygen vacancies. This notable increase suggests a more durable surface structure with the Ru_3_ cluster’s presence. Appendix A provide comparative data that elucidate this effect.

The findings, as depicted in Figure 4, demonstrate considerable alterations in the electronic structure of TiO_2_ resulting from its interaction with the Ru_3_ cluster and the occurrence of an oxygen vacancy. A key observation is the appearance of new energy states within the band gap, particularly hybrid states situated at −0.24 eV and −0.36 eV, corresponding to Ti_64_ and Ti_61_ atoms, respectively. These states are induced by the oxygen vacancy on the TiO_2_ surface. The HOMO of the Ru_3_ cluster is observed at a high energy level of approximately −0.24 eV. In addition, a distinct state at −0.58 eV is identified, corresponding to a polaron situated on a Ti_27_ atom, along with states localized on the Ru cluster itself. This polaronic state, characterized by an electron gain of about 0.3 e^−^ on the Ti_27_ atom, is illustrated in Figure 4. These localized states significantly enhance the photocatalytic performance of rutile TiO_2_ (110) when exposed to visible light irradiation. Therefore, the introduction of both the Ru_3_ cluster and oxygen vacancy on the TiO_2_ surface could be significant in reducing the energy required to catalyze water splitting.

This research uncovers that the Ru_3_ cluster contributes a noteworthy electron donation, around +0.69 e^−^, to the TiO_2_ surface. Compared to the structure lacking an oxygen vacancy (as depicted in Figure 2a), this electron transfer is diminished by 0.06 e^−^. The emergence of a polaronic state resulting from this electron transfer is instrumental in boosting the absorption of visible light photons, an essential factor for efficient photocatalysis. Notably, the combined effect of the Ru_3_ cluster and the oxygen vacancy on the TiO_2_ substrate markedly elevates its photocatalytic efficiency. This combined effect implies that the Ru_3_ cluster and the oxygen vacancy both act as potential catalysts in applications involving water splitting. The insights gained from this research are crucial for the enhancement of highly effective photocatalysts, especially those targeting photocatalytic hydrogen generation. The research highlights the significance of strategic interactions between metal clusters and substrates in enhancing photocatalytic systems and offers a roadmap for the methodical design of innovative materials in the field of renewable energy technologies.

Based on the detailed analysis of the interactions between Ru_3_ clusters and TiO_2_ rutile (110) surfaces for green hydrogen production, future research should focus on exploring the synergistic effects of other metal clusters similar to Ru_3_, such as those of platinum or palladium, to further enhance photocatalytic activity. In addition, comparative studies with single Ru atoms, Ru dimers, Ru_4_ clusters, and Ru nanoparticles on TiO_2_ would enrich our understanding of size-dependent catalytic effects. Additionally, investigating the influence of varying the surface morphology and defect density of TiO_2_ could provide deeper insights into optimizing photocatalytic efficiency. Furthermore, integrating experimental validation with the computational findings would be invaluable in advancing the practical applications of these systems in sustainable energy technologies. This could involve testing different cluster compositions and sizes on TiO_2_ surfaces under real-world conditions to assess their photocatalytic performance and durability. Future studies incorporating both DOS and band structure analyses would offer a more detailed insight into the electronic modifications induced by Ru_3_ loading and their impact on photocatalytic activity. This approach not only promises to validate and extend our current findings but also contributes to the broader understanding of material properties and their optimization for enhanced photocatalytic efficiencies, paving the way for significant advancements in material science and photocatalysis. The exploration of these avenues could lead to the development of more efficient and robust photocatalysts for green hydrogen production, contributing significantly to renewable energy solutions.

## 4. Concluding Remarks

The present study comprehensively investigates the interactions between Ru_3_ clusters and rutile TiO_2_ (110) surfaces, offering valuable insights into their photocatalytic behavior and potential applications in green hydrogen production. Utilizing advanced computational methods, i.e., DFT incorporated with HF theory, this research has explored the electronic and structural attributes of Ru_3_ clusters, both in isolation and when adsorbed onto pristine and defective TiO_2_ surfaces. The results underscore that clusters of Ru_3_ augment the photocatalytic efficiency of TiO_2_ by altering its electronic configuration and broadening its light absorption spectrum, notably within the visible light spectrum. This enhancement is caused by the introduction of new energy states, specifically the localized states, and an improved charge transfer mechanism, which are critical for efficient photocatalytic processes. Furthermore, this study delves into the stability of various Ru_3_ cluster configurations on TiO_2_ surfaces, underscoring the importance of cluster orientation and surface morphology in determining the photocatalytic efficiency. The most stable configurations for the adsorption of the Ru_3_ cluster led to a charge transfer of approximately +0.75 electrons to TiO_2_, causing the cluster to undergo oxidation. Furthermore, the inclusion of the Ru_3_ cluster on the TiO_2_ rutile (110) surface has been found to significantly enhance the material’s stability. This enhancement is quantitatively supported by the observation of a 0.38 eV increase in the energy required to form surface oxygen vacancies.

This study highlights the significance of Ru_3_ clusters in enhancing the photocatalytic efficiency of TiO_2_ for renewable energy and environmental remediation, particularly in hydrogen production from water. It offers insights into the electronic dynamics and stability of metal clusters on semiconductor surfaces, suggesting directions for developing new materials for sustainable energy. The approach combines DFT calculations and cluster-surface interaction analysis, setting a benchmark for future research. Further studies could explore various metal–semiconductor pairs and surface structures and combine experimental and computational methods to improve photocatalytic systems for real-world applications, contributing to the pursuit of green hydrogen and environmental sustainability.

## Figures and Tables

**Figure 1 nanomaterials-14-00396-f001:**
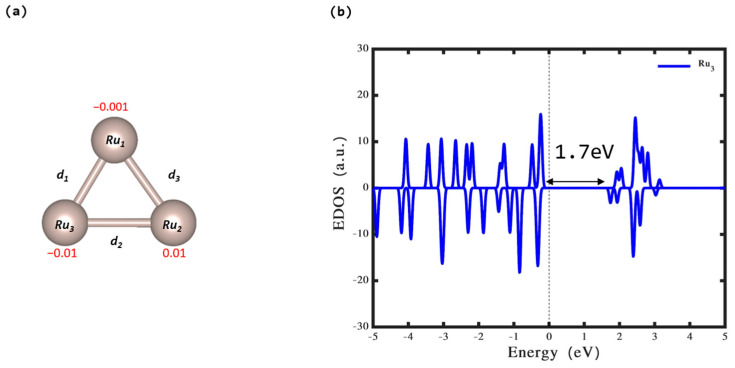
(**a**) The optimized structure of the Ru_3_ cluster and (**b**) its density of states (DOS). In the figure, red numerals indicate the electron count on each atom. The notations d_1_, d_2_, and d_3_ correspond to the lengths of the Ru-Ru bonds within the cluster. For detailed numerical data and specific measurements, refer to Table 1.

**Figure 2 nanomaterials-14-00396-f002:**
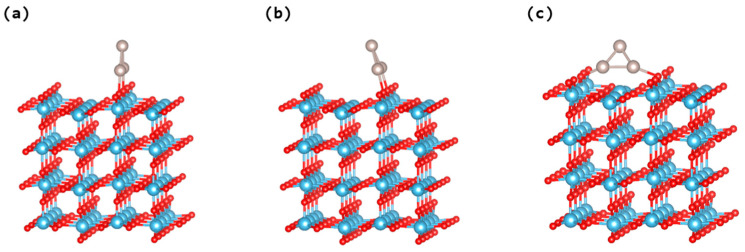
Various adsorption systems of triangle Ru_3_ clusters at rutile TiO_2_ (110) surface: (**a**) the upstanding cluster, (**b**) titled cluster (parallel to the bridging oxygen atoms), and (**c**) titled cluster (perpendicular to the bridging oxygen atoms). The Ru, O, and Ti atoms are shown by the silver, red, and blue circles.

**Figure 3 nanomaterials-14-00396-f003:**
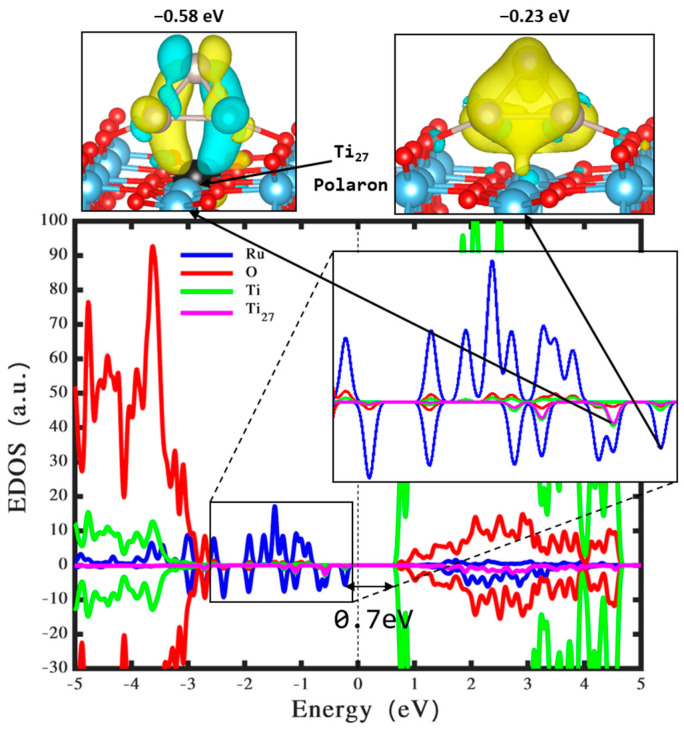
Density of states and the wavefunction of the most optimized Ru_3_ cluster on a perfect rutile TiO_2_ (110) surface. The states associated with Ti, O, Rh, and Ti_27_ atoms are indicated by green, red, blue, and pink colors, respectively. The Fermi energy level is indicated by the black vertical line. The reference colors yellow and blue for isosurfaces symbolize the positive and negative stages of wave functions, respectively. It is important to note that these reference colors are consistently used for all wavefunction plots in the following figures.

**Figure 4 nanomaterials-14-00396-f004:**
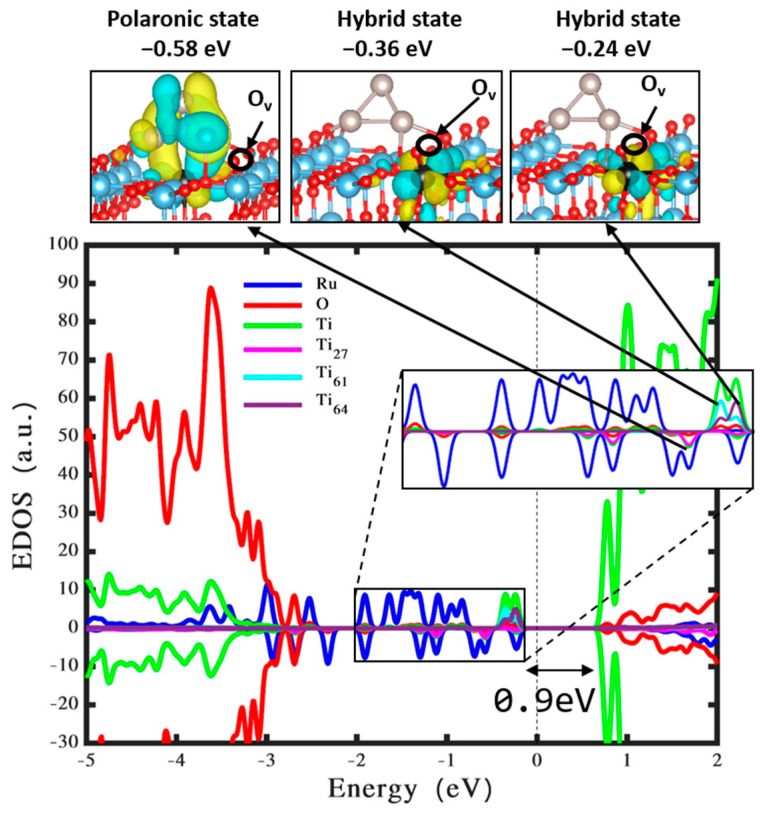
Density of states and the wavefunction of the most stable Ru_3_ cluster positioned on a defective rutile TiO_2_ (110). The states corresponding to Rh, O, Ti, Ti_27_, Ti_61_, and Ti_64_ atoms are signified by blue, red, green, pink, cyan, and purple colors, respectively.

**Table 1 nanomaterials-14-00396-t001:** Bond lengths for the Ru_3_ cluster as depicted in Figure 1a.

Bond Length (Å)	Ru_3_
d_1_	2.20
d_2_	2.33
d_3_	2.48

**Table 2 nanomaterials-14-00396-t002:** Adsorption energies ($E_{ads}$) calculated using DFT + U and the Bader charge distributions for the trapezoidal Ru_3_ clusters, as illustrated in Figure 2.

Structure	Figure 2a	Figure 2b	Figure 2c
*E_ads_* (eV)	−4.04	−4.00	−5.15
Charge on Ru_3_ (e^−^)	+0.78	+0.76	+0.75

## Data Availability

Data are contained within this article.

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
