# Peer review of "Geometrical Stabilities and Electronic Structures of Ru3 Clusters on Rutile TiO2 for Green Hydrogen Production"

_nanomaterials, 2024, doi:10.3390/nano14050396_

Round 1

Reviewer 1 Report

Comments and Suggestions for Authors

Authors calculated three different cases for loading of Ru3 cluster on TiO2 by DFT and revealed that the improvement of photocatalytic performance of Ru-TiO2 was attributed to the alteration of electronic configuration and broadening of visible light absorption spectrum. This work is of certain significance. I suggested its publication in Nanomaterials after revision.

1. How does Ru3 cluster framework stabilize on TiO2? The relative data need to be provided compared with Ru and dimer or Ru4.

2. How to explain the relationship between the broadening of adsorbed visible light spectrum and energy required for water splitting (H2 and O2 production). Not all materials that can absorb visible light are suitable for catalyzing water splitting.  

Comments on the Quality of English Language

Minor editing of English language required.

Reviewer 2 Report

Comments and Suggestions for Authors

The present work reported a theoretical study on the geometry and electronic properties of Ru3 clusters anchored to rutile TiO2 (110) surfaces using DFT and HSE06 hybrid functional calculations. The obtained results indicate that introducing these clusters alters the electronic structure of TiO2 by introducing localized states in the bandgap, which leads to the broadening of the light adsorption to the visible region. There are critical issues in this work that need to be carefully revised. Otherwise, I would not recommend this work for publication.

1. The title of this work should be revised as it does not reflect the novelty of this work. It is too general and looks like the title of a review article, especially with the phrase “a route to green hydrogen”.

     1. Calculated band structure of pristine TiO2 and Ru3-loaded TiO2 should be provided, in addition to the density of states (DOS), to confirm the bandgap modification.

     2. In this work, the authors desire to demonstrate the enhanced photocatalytic activity of TiO2 toward the water splitting upon the loading of Ru3 clusters as co-catalysts. However, they only showed modifications of the TiO2 electronic structure, which leads to improved light adsorption. Meanwhile, one of the most essential aspects – the activity of Ru clusters toward the hydrogen evolution reaction (HER) was not investigated. It should be noted that enhanced light absorption does not mean improved photocatalytic activity. The authors must calculate the Gibbs free energy of H adsorption GH) of Ru clusters and compare it to that of pristine. This is an essential quantitative benchmark to determine HER efficiency. On the other hand, there are various possible active sites for HER on the surface of Ru3-loaded TiO2, including Ru3 cluster, surface O atoms, surface Ti atoms, Ti3+ species (formed upon the formation of oxygen vacancies), Ti27, Ti61, etc., whose HER activity must be investigated. If Ru3 clusters are not active or slightly for HER, they are not co-catalysts but may boost the catalytic activity of TiO2 active sites.

   3. To verify the enhancement of catalytic activity by loading Ru3 clusters, some referent samples must be employed for the comparison. Single Ru atoms and Ru nanoparticles could be good candidates.

     4. The conclusion should be revised. The summary of this work (the first paragraph) should be more detailed, while the second paragraph addressing the general interest of this work should be shortened.   

Round 2

Reviewer 1 Report

Comments and Suggestions for Authors

Authors have slightly improved the quality of manuscript. Due to some special causes, some calculations what I want to see were not shown. 

Reviewer 2 Report

Comments and Suggestions for Authors

The authors have thoroughly revised the manuscript and provided appropriate explanations. I therefore recommend publication of this work in the present form.